# Aspirin Is Related to Worse Clinical Outcomes of COVID-19

**DOI:** 10.3390/medicina57090931

**Published:** 2021-09-04

**Authors:** Isaac Kim, Siyeong Yoon, Minsup Kim, Hyunil Lee, Sinhyung Park, Wonsang Kim, Soonchul Lee

**Affiliations:** 1Department of General Surgery, CHA Bundang Medical Center, CHA University School of Medicine, Seongnam 13488, Korea; isaac24@cha.ac.kr; 2Department of Orthopedic Surgery, CHA Bundang Medical Center, CHA University School of Medicine, Seongnam 13488, Korea; tldud1105@naver.com; 3inCerebro Drug Discovery Institute, Seoul Technopark, Seoul 01811, Korea; minsupkim@incerebro.com (M.K.); wonsangkim@incerebro.com (W.K.); 4Department of Orthopedic Surgery, Ilsan Paik Hospital, Inje University, Goyang 10380, Korea; hyunil.lee7@gmail.com; 5Department of Orthopedic Surgery, Bucheon Hospital, Soonchunhyang University, Bucheon 14584, Korea; greatpsh78@gmail.com

**Keywords:** COVID-19, aspirin, outcome

## Abstract

*Background**and Objectives*: Aspirin is used globally to reduce pain and inflammation; however, its effect in patients with coronavirus disease (COVID-19) is not fully investigated and remains controversial. We evaluated the association between aspirin and COVID-19 outcomes using nationwide data from the Korean National Health Insurance System. *Materials and Methods:* This was a retrospective observational cohort study that included 22,660 eligible patients who underwent COVID-19 testing in South Korea between 1 January–31 July 2020. We identified all aspirin users prescribed aspirin within two weeks before or after the index date. The primary outcome was positivity for the COVID-19 test, and secondary outcomes included conventional oxygen therapy, intensive care unit, mechanical ventilation, or death. We applied the propensity score matching method to reduce the possible bias originating from the differences in patients’ baseline characteristics. *Results:* Of those eligible, 662 patients were prescribed aspirin. Among them, 136 patients were on aspirin within two weeks before diagnosis and 526 patients were on aspirin after diagnosis. The COVID-19 test positivity rate was not significantly different according to aspirin use. Aspirin use before COVID-19 was related to an increased death rate and aspirin use after COVID-19 was related to a higher risk of the conventional oxygen therapy. *Conclusion:* Aspirin use was associated with adverse effects in COVID-19 patients. Further studies for mechanisms are needed.

## 1. Introduction

A major outbreak of coronavirus disease (COVID-19) caused by severe acute respiratory syndrome coronavirus 2 (SARS-CoV-2) in Wuhan, China, was first reported in December 2019. More than 144,000 cases of COVID-19 and approximately 2000 deaths were reported throughout Korea between January 2020 and June 2021 [1]. Globally, to date, more than 100 million cases of COVID-19 and over 3 million deaths have been reported [2].

In patients hospitalized with COVID-19, dexamethasone is associated with lower mortality [3]; however, no absolute medical treatment has been shown to improve mortality in patients with COVID-19. Among several treatments implemented for COVID-19, aspirin deserves attention because it is inexpensive, widely available, and has a well-described risk profile. Aspirin was first synthesized in 1898, and it has been used to treat and prevent various diseases. Aspirin has anti-inflammatory, analgesic, antipyretic, and antithrombotic effects [4]. Although the exact mechanism is unclear, it can be assumed that aspirin has anti-thrombotic, anti-inflammatory, and immune-modulatory effects [5]. In fact, aspirin is known to be associated with reduced mortality and reduced risk of acute respiratory distress syndrome (ARDS) in patients with severe COVID-19 [6]. In addition, Chow et al. reported that hospitalized COVID-19 patients with low-dose aspirin daily use for protection against cardiovascular diseases had a significantly lower risk of complications and mortality, compared with patients who did not receive the aspirin [7]. On the other hand, Yuan et al. reported that the use of low-dose aspirin was not related with the clinical outcomes of COVID-19 patients with coronary artery disease [8]. The use of aspirin is controversial, and the related research is limited. Moreover, there was no study which analyzed the effectiveness of aspirin based on the time it was taken.

In this study, we used nationwide COVID-19 data from the Korean National Health Insurance System (NHIS) to investigate the association between aspirin use and mortality. In particular, we subdivided patients and analyzed the use of aspirin before and after the diagnosis of COVID-19.

## 2. Materials and Methods

### 2.1. Data Source

The NHIS-COVID-19 cohort database was developed for medical research purposes in cooperation with the Health Insurance Review & Assessment of Korea (HIRA) and the Korea Centers for Disease Control & Prevention (KCDC). This cohort database provided data of all patients who underwent COVID-19 testing in South Korea between 1 January 2020 and 31 July 2020 (*N* = 25,739). Data included information on SARS-CoV-2 testing date, treatment results, and demographic information. The NHIS-COVID-19 database comprised three groups: COVID-19-positive patients, COVID-19-negative patients, and control groups. COVID-19-negative patients were individuals who tested negative for COVID-19, and control groups comprised 15 times the number of COVID-19-positive patients using stratification methods of sex, age, and place of residence. The NHIS-COVID-19 database included disease diagnoses according to the 10th revision of the International Statistical Classification of Diseases (ICD-10) codes, prescription information concerning drugs, COVID-19-related outcomes, and death records from 2015 to 2020. All personal data used in our study were de-identified to ensure confidentiality.

### 2.2. Study Population

We defined the date of the first COVID-19 test for each patient as the index date of the cohort. We included all patients who underwent COVID-19 testing between 1 January 2020 and 31 July 2020 and excluded patients if they were younger than 20 years (*N* = 3079); finally, a total of 22,660 patients were included in the base cohort (Figure 1). We extracted and combined patient’s characteristic information such as sex, age (20–29, 30–39, 40–49, 50–59, 60–69, 70–79, 80+), and region of residence (Seoul, Gyeonggi, Daegu, Gyeongbuk, and others). History of underlying diseases (such as hypertension [HTN], chronic obstructive pulmonary disease [COPD], asthma, chronic kidney disease [CKD], diabetes mellitus [DM], and cerebrovascular disease [CVD]) was confirmed by assessment of at least two claims within 1 year, using the appropriate ICD-10 codes (Appendix A). The Charlson Comorbidity Index score (using ICD-10 codes) was calculated using previously reported methods [1]. The region of residence was classified as Seoul, Gyeonggi, Daegu, Gyeongbuk, and others. Current use of systemic steroids was defined as consuming drugs within 30 days before the index date, and the types and codes for steroids were referenced to a previous study [9].

### 2.3. Exposure

We identified all aspirin users who were prescribed aspirin within two weeks before or after the index date. Aspirin was classified as Anatomical Therapeutic Chemical (ATC) codes B01AC06 and N02BA01. Non-users were defined as patients who had not received aspirin within two weeks before or after the index date.

### 2.4. Outcomes

The positivity of laboratory test results for COVID-19 was defined as the primary outcome. The secondary outcomes included composite endpoint 1 (conventional oxygen therapy, intensive care unit, mechanical ventilation, or death) and composite endpoint 2 (intensive care unit, mechanical ventilation, or death).

### 2.5. Statistical Analysis

Statistical significance was set at *p* < 0.05. Data were analyzed using the Pearson χ2 test or Fisher’s exact test for categorical variables. We applied the propensity score matching (PSM) method to reduce the possible bias originating from the differences in patients’ baseline characteristics. We performed a logistic regression model with adjustments for the following: sex, age (20–29, 30–39, 40–49, 50–59, 60–69, 70–79, 80+), region of residence (Seoul, Gyeonggi, Daegu, Gyeongbuk, and others), HTN, COPD, asthma, CKD, DM, CVD, Charlson Comorbidity Index (0, 1, or ≥ 2), and current use of systemic steroids. We assessed the PSM of the two groups on the basis of a greedy algorithm of nearest neighbor matching at a 1:1 fixed ratio and estimated the predicted probability of (1) aspirin users before the COVID-19 index date versus non-aspirin users among all patients who underwent COVID-19 testing (*N* = 26,660); (2) aspirin users after the COVID-19 index date versus non-aspirin users among patients who underwent COVID-19 testing. We examined the balance of covariate distribution between the two groups using standardized mean differences (SMDs), which is the most commonly used statistical method that allows comparison between variables with different units of measurement [10]. We identified certain clinical endpoints of COVID-19 among patients with confirmed COVID-19. The exposure was before or after the use of aspirin based on the COVID-19 index date. The primary endpoint was a positive SARS-CoV-2 test result. The secondary endpoints were composite endpoint and severe clinical outcomes. Data manipulation and statistical analyses were performed using R software (version 3.6.3; The R Foundation for Statistical Computing, Vienna, Austria; http://www.R-project.org/ accessed on 8 April 2021).

## 3. Results

According to the data from the HIRA between 1 January 2020 and 31 July 2020, a total of 26,660 patients were diagnosed with COVID-19. Of these, 21,998 patients were not prescribed aspirin and 662 patients were prescribed aspirin. Among them, 136 patients were on aspirin within 2 weeks before diagnosis and 526 patients were administered aspirin after diagnosis. The baseline characteristics of the patients are shown in Table 1. In the entire cohort, 12,668 (55.9%) patients were women and women accounted for the majority of non-aspirin users, while men accounted for the majority of patients who were aspirin users. Regarding age, patients in their 20s were most common in the overall cohort; however, patients aged 70 years or older were the most common aspirin users. Hypertension was the most common comorbidity.

In the two cohorts, patients who took aspirin before the COVID-19 diagnosis (*n* = 136) and those who took aspirin after COVID-19 (*n* = 526) were matched individually to an equal number of non-aspirin-exposed patients in our two propensity score-matched cohorts (Table 2). No major imbalances in the demographics and clinical characteristics were observed when evaluated using SMD within groups in the propensity-matched cohorts (SMD < 0.25). In patients who had aspirin before COVID-19 diagnosis, minimally adjusted odds (accounting for age and gender via PSM) of COVID-19 positivity were 0.71 (95% CI, 0.34–1.46) and fully adjusted odds (accounting for age, gender, region of residence, comorbidities, and current use of steroids) were 0.63 (95% CI, 0.27–1.45) (Figure 2).

The COVID-19 test positivity rates were not significantly different between the groups (Table 3). The positivity rate in patients without a history of aspirin use was 14.7% (20/136), compared with 11% (15/136) in those who took aspirin before COVID-19 (*p*-value = 0.469). Aspirin use before COVID-19 was related to an increased risk of the composite endpoint of COVID-19 (Table 4). The rate of composite endpoint 1 was 80% (12/15) for aspirin use before COVID-19 and 40% (8/20) in the non-aspirin group (*p*-value = 0.043). The rate of composite endpoint 2 was also increased in patients who took aspirin before COVID-19 compared to that in the non-aspirin group (60% versus 15%, *p*-value = 0.016). However, sub-analysis according to each factor of composite endpoint showed different results, wherein the death rate alone was significantly different between the non-aspirin and aspirin group before COVID-19 diagnosis (Table 4, 5% vs. 40%, *p* = 0.027). 

The positivity rate in the non-aspirin group was 25.7% (135/526) compared with 23.6% (124/526) in the aspirin group after COVID-19 diagnosis (*p*-value = 0.474). Aspirin use after COVID-19 diagnosis was related to an increased risk of the composite endpoint 1 (Table 5). The rate of composite endpoint 1 was 45.2% (56/124) in the aspirin use after COVID-19 diagnosis group and 31.9% (43/135) in the non-aspirin group (*p*-value = 0.038), whereas the rate of composite endpoint 2 was not statistically different between the two groups (21.5% versus 17.7%, *p*-value = 0.549). In the sub-analysis according to each factor of the composite endpoint, only the conventional oxygen therapy rate was significantly increased in the aspirin after COVID-19 group compared to that in the non-aspirin group (Table 5, 46.7% versus 35.0%, *p*-value < 0.0001).

## 4. Discussion

In this Korean nationwide cohort, we investigated whether aspirin use decreased susceptibility to COVID-19 among 22,660 patients who underwent COVID-19 testing. We found that aspirin use, both before and after the diagnosis of COVID-19, did not decrease susceptibility to COVID-19 infection. Meanwhile, aspirin use before COVID-19 was related to an increased risk of worsened outcomes, especially death, and aspirin use after COVID-19 was related to an increased rate of conventional oxygen therapy.

In previous studies, the role of aspirin in patients with COVID-19 is controversial. Theoretically, as severe COVID-19 infection is mainly a multisystem inflammatory process, the use of aspirin can provide positive outcomes. In fact, several studies have explained the benefits of aspirin use among COVID-19 patients. 

Paranjpe et al. showed that systemic anticoagulation could reduce mortality in mechanically ventilated COVID-19 patients [11] and Ikonomidis et al. showed that aspirin could reduce the incidence of cytokine storm in patients with CVD and COVID-19 [12]. They elucidated that aspirin decreases the production of interleukin-6 (IL-6), C-reactive protein (CRP), and macrophage colony-stimulating factor [13]. Klok et al. reported that COVID-19 is associated with hypercoagulability and pulmonary microthrombosis, and aspirin may mitigate these effects [14]. In an autopsy study, COVID-19 infection was shown to be associated with pulmonary embolism and alveolar micro-thrombi [15]. COVID-19 also causes vascular endothelialitis involving the pulmonary capillary endothelium [16] and infects endothelial cells in multiple organs [16,17]. According to Warner et al., aspirin is an inhibitor of COX-1 and it decreases thromboxane A2 synthesis, platelet aggregation, and thrombus formation [4]. In another study, aspirin’s potential benefits in lung injury are thought to be related to reduced platelet-neutrophil aggregates in the lungs, reduced inflammation, and increased lipoxin formation, which restores pulmonary endothelial cell function [18]. These findings may explain why aspirin has a beneficial effect in COVID-19 patients. 

In contrast, some researchers have argued that aspirin has harmful effects on COVID-19 patients. The Belgian Federal Agency for Medicines and Health Products released a statement on 16 March 2020, stating that nonsteroidal anti-inflammatory drugs (NSAIDs) can lead to serious complications in COVID-19 patients [19]. In a rat model study, NSAIDs possibly increased the expression of ACE2 [20], a pivotal receptor of COVID-19, which potentially increases COVID-19 infectivity. In a clinical cohort study, there was no difference in aspirin usage on the survival and non-survival of COVID-19 patients [21]. According to a recent meta-analysis on the effect of aspirin on COVID-19 patients, there is no association between the use of aspirin and mortality in patients with COVID-19 [22]. Although patients on aspirin tend to have higher risk factors for severe COVID-19 infection (e.g., older age, pre-existing coronary artery disease, diabetes mellitus, etc.), they suggest no protective effect of aspirin among different groups of patients. Three studies were included in the meta-analysis. First, Yuan et al. analyzed whether pre-hospitalization use of aspirin was associated with mortality in COVID-19 patients with coronary artery disease (CAD) [8]. They recruited 183 patients with CAD in China (52 patients with aspirin use and 131 without aspirin use) and showed that the use of aspirin was not correlated with mortality in multivariate analysis (OR = 0.944, 95% CI: 0.411–2.171, *p*-value = 0.893). Although this study was the same as ours in that it included Asian patients, it was different in that only patients with CAD were included in analysis. Second, Chow et al. retrospectively analyzed whether aspirin use was associated with a reduced risk of mechanical ventilation, intensive care unit admission, and mortality in the United States [7]. In that study, 314 patients without aspirin use and 98 patients with aspirin were included, which showed that aspirin use may be associated with improved outcomes in hospitalized COVID-19 patients. In particular, mechanical ventilation (48.4% vs. 35.7%, *p*-value = 0.03) and intensive care unit admission rates (51% vs. 38.8%, *p*-value = 0.04) were higher in patients who did not use aspirin. Since there were more patients with underlying diseases (e.g., HTN, DM, CAD, renal disease, and liver disease) in the aspirin group, the results could be different after PSM for even distribution with underlying disease. Lastly, Alamdari et al. conducted a retrospective study of prognostic factors associated with mortality in COVID-19 patients in Iran [21]. They analyzed 396 surviving and 63 non-surviving patients, and showed that the fatality rate was markedly higher among patients with comorbidities such as diabetes (*p*-value = 0.018), malignancy (*p*-value = 0.008), CKD (*p*-value = 0.002), and body mass index (BMI) > 35 (*p*-value = 0.0003). The use of aspirin was higher in the non-surviving patients group (11.1% vs. 14.3%); however, the difference was not significant (*p*-value = 0.52). If the comorbidity ratio was evenly matched with PSM, after which the mortality rate was calculated according to aspirin usage, the results might have been different. 

In our study, aspirin use before COVID-19 was related to an increased risk of the composite endpoint, but death rate alone was significantly different between the non-aspirin and aspirin group before COVID-19. Aspirin use after COVID-19 was related to an increased risk of conventional oxygen therapy. We could not find enough evidence to support our research, owing to a lack of studies on the effects of aspirin in COVID-19 patients. We tried to find more evidence in studies on the effect of aspirin on ARDS. ARDS develops in 42% of patients with COVID-19 pneumonia [23]. 

According to a meta-analysis on the effects of aspirin on ARDS [18], some studies have demonstrated benefits, while others have reported otherwise. Of the 15 preclinical trials, 13 reported a beneficial effect of aspirin on ARDS, evidenced by improved oxygenation, diminished lung edema, inflammation, and increased survival. On the other hand, no benefit was observed in one study that conversely demonstrated the worsening of inflammation [24]. There was a multicenter, double-blind, placebo-controlled, randomized clinical trial to evaluate the efficacy and safety of early aspirin administration for the prevention of ARDS [25]. They examined 390 patients at high risk for ARDS who were randomized to aspirin or placebo and found that aspirin did not prevent ARDS at 7 days (10.3% vs. 8.7%, respectively; odds ratio, 1.24 (92.6% CI, 0.67–2.31); *p*-value = 0.53) or improve 28-day survival (90% vs. 90%, respectively), with hazard ratio (90% CI) 1.03 (0.60–1.79); *p*-value = 0.92. 

Meanwhile, there was a recent study for the correlation between pre-diagnosis aspirin and mortality in COVID-19 patients in the United States [26]. The study included 6842 patients with aspirin and 21508 patients without aspirin, and they were matched using propensity scores. They showed that preexisting aspirin prescription was associated with a statistically and clinically significant decrease in overall mortality at 14 days (OR 0.38, 95% CI, 0.32–0.46) and at 30 days. (OR 0.38, 95% CI, 0.33–0.45).

We could not find sufficient evidence to explain why the results of other studies were different from our results. The results may vary depending on the race and research method used. In addition, we could not establish a direct cause or effect of aspirin on COVID-19 patients through a retrospective evaluation. However, based on previous data, it could be assumed that many patients who used aspirin already had CAD, and the mortality of COVID-19 was higher in patients with CAD [27]. Therefore, aspirin use before COVID-19 was related to an increased risk of the composite endpoint. Another study showed that aspirin use increased IL-2 concentrations after treatment with aspirin on day 1 [28]. In a recent study on immune-based biomarkers to predict clinical outcomes in COVID-19 patients, IL-2 was one of the 12 biomarkers whose increased levels were associated with increased mortality [29]. Hence, we could assume that IL-2 may be a factor in the adverse effects of aspirin in patients with aspirin after COVID-19, and further studies are needed to validate this. 

This study had several advantages. Importantly, our data source included a large number of Korean National Health Insurance System data, and we used PSM to rule out the effects of factors other than aspirin. In addition, this is the first study on the effect of aspirin in Asian COVID-19 patients using PSM and a unique analysis of aspirin use before and after COVID-19 diagnosis. 

The results of this study should be interpreted considering the following limitations. First, this was a retrospective study, and despite our efforts to adjust all confounding factors by PSM analysis, unmeasured factors might have affected the results. Second, the database used in this study retrieved information from the Korean National Health Insurance System; therefore, clinical presentation, symptoms, hospital course, and biomarker data could not be included. Third, we analyzed only prescribed data for aspirin use, although aspirin is over-the-counter medicine in Korea. Fourth, some patients who were experiencing COVID-19 but not diagnosed could be included in the “aspirin before COVID-19” cohort. Lastly, the results are derived from a cohort of Koreans, and the results might be different in other countries. Despite these limitations, this study provides real-world data based on the effects of aspirin use in COVID-19 patients. 

## 5. Conclusions

Aspirin use before the diagnosis of COVID-19 tended to increase the death rate, and aspirin use after the diagnosis of COVID-19 tended to increase the conventional oxygen therapy rate, compared to no use. Analysis with a larger sample size is required to confirm the adverse effects of aspirin on COVID-19, and further research for mechanisms is needed.

## Figures and Tables

**Figure 1 medicina-57-00931-f001:**
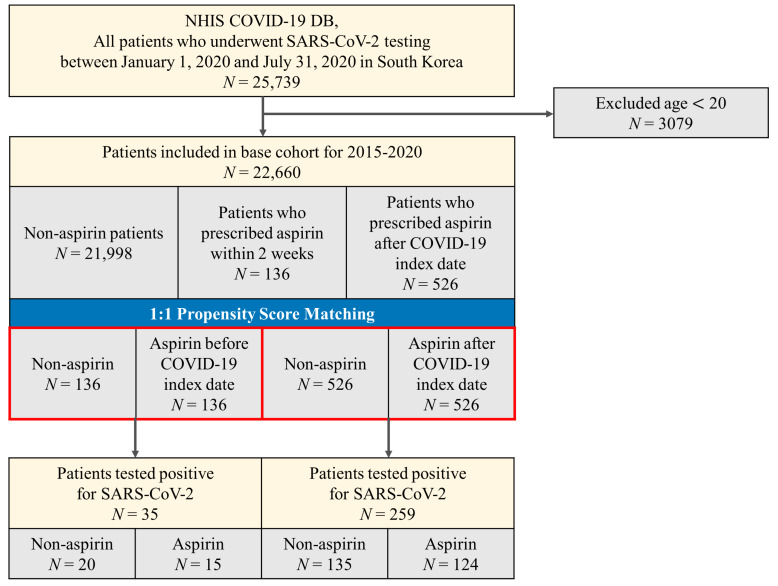
Disposition of patients in the Korean nationwide cohort.

**Figure 2 medicina-57-00931-f002:**
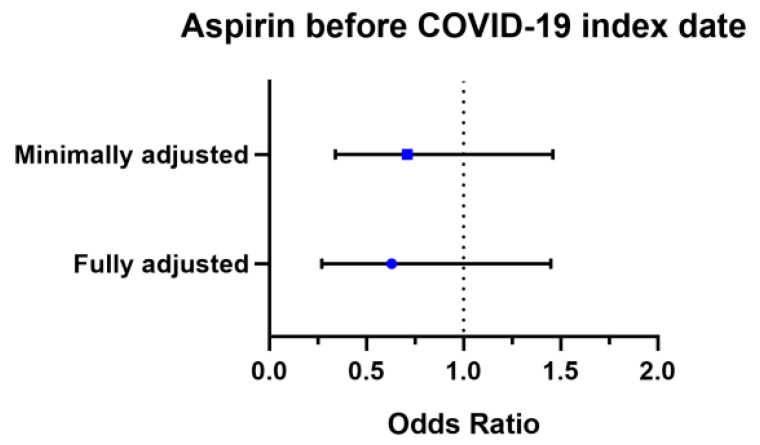
Forest plot of the association between aspirin and COVID-19 test positivity.

**Table 1 medicina-57-00931-t001:** Baseline characteristics of patients who were prescribed Aspirin before or after COVID-19 index date in NHIS COVID-19 database.

Characteristic	Entire Cohort *N* = 22,660	Non-Aspirin *N* = 21,998	Aspirin Before COVID-19 Index Date *N* = 136	Aspirin After COVID-19 Index Date *N* = 526
Sex, *n* (%)				
Male	9992 (44.1)	9601 (43.6)	86 (63.2)	305 (58.0)
Female	12,668 (55.9)	12,397 (56.4)	50 (36.8)	221 (42.0)
Age, *n* (%)				
20–29	4841 (21.4)	4835 (22.0)	0 (0.0)	6 (1.1)
30–39	3764 (16.6)	3758 (17.1)	1 (0.7)	5 (1.0)
40–49	3399 (15.0)	3374 (15.3)	7 (5.1)	18 (3.4)
50–59	3662 (16.2)	3598 (16.4)	17 (12.5)	47 (8.9)
60–69	3068 (13.5)	2914 (13.2)	36 (26.5)	118 (22.4)
70–79	2156 (9.5)	1945 (8.8)	47 (34.6)	164 (31.2)
80+	1770 (7.8)	1574 (7.2)	28 (20.6)	168 (31.9)
Region, *n* (%)				
Seoul	3902 (17.2)	3794 (17.2)	22 (16.2)	86 (16.3)
Gyeonggi	2726 (12.0)	2637 (12.0)	17 (12.5)	72 (13.7)
Daegu	8592 (37.9)	8401 (38.2)	36 (26.5)	155 (29.5)
Gyeongbuk	1831 (8.1)	1759 (8.0)	14 (10.3)	58 (11.0)
Others	5609 (24.8)	5407 (24.6)	47 (34.6)	155 (29.5)
HTN, *n* (%)	5675 (25.0)	5170 (23.5)	108 (79.4)	397 (75.5)
COPD, *n* (%)	767 (3.4)	703 (3.2)	13 (9.6)	51 (9.7)
Asthma, *n* (%)	2433 (10.7)	2314 (10.5)	23 (16.9)	96 (18.3)
CKD, *n* (%)	1071 (4.7)	895 (4.1)	62 (45.6)	114 (21.7)
DM, *n* (%)	3713 (16.4)	3354 (15.2)	92 (67.6)	267 (50.8)
CVD, *n* (%)	1888 (8.3)	1631 (7.4)	66 (48.5)	191 (36.3)
Charlson Comorbidity Index, *n* (%)				
0	11,775 (52.0)	11,719 (53.3)	12 (8.8)	44 (8.4)
1	3029 (13.4)	2861 (13.0)	35 (25.7)	133 (25.3)
2 or more	7856 (34.7)	7418 (33.7)	89 (65.4)	349 (66.3)
Current use of medication, *n* (%)				
Steroid	1921 (8.5)	1826 (8.3)	36 (26.5)	59 (11.2)

HTN: Hypertension; COPD: Chronic obstructive pulmonary disease; CKD: Chronic kidney disease; DM: Diabetes mellitus; CVD: Cerebrovascular disease.

**Table 2 medicina-57-00931-t002:** Propensity score-matched baseline characteristics and COVID-19 test positivity between non-aspirin and aspirin groups.

Characteristic	Non-Aspirin *N* = 136	Aspirin before COVID-19 Index Date *N* = 136	SMD	Non-Aspirin *N* = 526	Aspirin after COVID-19 Index Date *N* = 526	SMD
Sex, *n* (%)			0.030			0.023
Male	84 (61.8)	86 (63.2)		299 (56.8)	305 (58.0)	
Female	52 (38.2)	50 (36.8)		227 (43.2)	221 (42.0)	
Age, *n* (%)			0.128			0.052
20–29	0 (0.0)	0 (0.0)		3 (0.6)	6 (1.1)	
30–39	1 (0.7)	1 (0.7)		7 (1.3)	5 (1.0)	
40–49	6 (4.4)	7 (5.1)		17 (3.2)	18 (3.4)	
50–59	14 (10.3)	17 (12.5)		41 (7.8)	47 (8.9)	
60–69	30 (22.1)	36 (26.5)		118 (22.4)	118 (22.4)	
70–79	52 (38.2)	47 (34.6)		160 (30.4)	164 (31.2)	
80+	33 (24.3)	28 (20.6)		180 (34.2)	168 (31.9)	
Region, *n* (%)			0.027			0.006
Seoul	17 (12.5)	22 (16.2)		93 (17.7)	86 (16.3)	
Gyeonggi	41 (30.1)	36 (26.5)		158 (30.0)	155 (29.5)	
Daegu	17 (12.5)	17 (12.5)		63 (12.0)	72 (13.7)	
Gyeongbuk	13 (9.6)	14 (10.3)		54 (10.3)	58 (11.0)	
Others	48 (35.3)	47 (34.6)		158 (30.0)	155 (29.5)	
HTN, *n* (%)	112 (82.4)	108 (79.4)	0.072	401 (76.2)	397 (75.5)	0.018
COPD, *n* (%)	11 (8.1)	13 (9.6)	0.050	38 (7.2)	51 (9.7)	0.083
Asthma, *n* (%)	26 (19.1)	23 (16.9)	0.059	85 (16.2)	96 (18.3)	0.054
CKD, *n* (%)	54 (39.7)	62 (45.6)	0.118	93 (17.7)	114 (21.7)	0.097
DM, *n* (%)	98 (72.1)	92 (67.6)	0.094	262 (49.8)	267 (50.8)	0.019
CVD, *n* (%)	67 (49.3)	66 (48.5)	0.015	184 (35.0)	191 (36.3)	0.028
Charlson Comorbidity Index, *n* (%)			0.113			0.003
0	11 (8.1)	12 (8.8)		42 (8.0)	44 (8.4)	
1	27 (19.9)	35 (25.7)		136 (35.9)	133 (25.3)	
2 or more	98 (72.1)	89 (65.4)		348 (66.2)	349 (66.3)	
Current use of medication, *n* (%)						
Steroid	35 (25.7)	36 (26.5)	0.017	54 (10.3)	59 (11.2)	0.030
COVID-19, *n* (%)						
Minimally adjusted OR *	1.00 (reference)	0.71 (0.34–1.46)		1.00 (reference)	0.89 (0.67–1.18)	
Fully adjusted OR †	1.00 (reference)	0.63 (0.27–1.45)		1.00 (reference)	0.89 (0.64–1.24)	

HTN: hypertension, COPD: Chronic obstructive pulmonary disease, CKD: Chronic kidney disease, DM: Diabetes mellitus, CVD: Cerebrovascular disease; * Minimally adjusted: adjustment for age and sex; † Fully adjusted: adjustment for age, sex, and region of residence (Seoul, Gyeonggi, Daegu, Gyeongbuk, others), HTN, COPD, Asthma, CKD, DM, CVD, Charlson Comorbidity Index (0, 1, 2 or more), and current use of steroid; SMD: standardized mean difference.

**Table 3 medicina-57-00931-t003:** Difference of COVID-19 diagnosis between non-aspirin and aspirin users before COVID-19 index date.

Variables	Non-Aspirin *N* = 136	Aspirin before COVID-19 Index Date *N* = 136	*p*-Value *
COVID-19, *n* (%)			0.469
No	116 (85.3)	121 (89.0)	
Yes	20 (14.7)	15 (11.0)	

* Pearson’s chi-square test.

**Table 4 medicina-57-00931-t004:** Clinical outcomes of COVID-19 between non-aspirin and aspirin users before COVID-19 index date.

Variables	Non-Aspirin *N* = 20	Aspirin before COVID-19 Index Date *N* = 15	*p*-Value
Composite endpoint 1, *n* (%)			0.043 *
No	12 (60.0)	3 (20.0)	
Yes	8 (40.0)	12 (80.0)	
Composite endpoint 2, *n* (%)			0.016 *
No	17 (85.0)	6 (40.0)	
Yes	3 (15.0)	9 (60.0)	
Conventional oxygen therapy			0.727 *
No	13 (65.0)	8 (53.3)	
Yes	7 (35.0)	7 (46.7)	
Intensive care unit			0.141 †
No	19 (95.0)	11 (73.3)	
Yes	1 (5.0)	4 (26.7)	
Mechanical ventilation			0.141 †
No	19 (95.0)	11 (73.3)	
Yes	1 (5.0)	4 (26.7)	
Death			0.027 †
No	19 (95.0)	9 (60.0)	
Yes	1 (5.0)	6 (40.0)	

* Pearson’s chi-square test; † Fisher’s exact test; Composite endpoint 1: conventional oxygen therapy, intensive care unit, mechanical ventilation, or death; Composite endpoint 2: intensive care unit, mechanical ventilation, or death.

**Table 5 medicina-57-00931-t005:** Clinical outcomes of COVID-19 between non-aspirin and aspirin users after COVID-19 index date.

Variables	Non-Aspirin *N* = 135	Aspirin after COVID-19 Index Date *N* = 124	*p*-Value *
Composite endpoint 1, n (%)			0.038
No	92 (68.1)	68 (54.8)	
Yes	43 (31.9)	56 (45.2)	
Composite endpoint 2, *n* (%)			0.549
No	106 (78.5)	102 (82.3)	
Yes	29 (21.5)	22 (17.7)	
Conventional oxygen therapy			0.000
No	108 (80.0)	73 (58.9)	
Yes	7 (35.0)	7 (46.7)	
Intensive care unit			0.447
No	131 (97.0)	117 (94.4)	
Yes	4 (3.0)	7 (5.6)	
Mechanical ventilation			0.173
No	128 (94.8)	111 (89.5)	
Yes	7 (5.2)	13 (10.5)	
Death			0.253
No	112 (83.0)	110 (88.7)	
Yes	23 (17.0)	14 (11.3)	

* Pearson’s chi-square test; Composite endpoint 1: conventional oxygen therapy, intensive care unit, mechanical ventilation, or death; Composite endpoint 2: intensive care unit, mechanical ventilation, or death.

## Data Availability

Data is contained within the article.

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
