# Peer review of "Aspirin Is Related to Worse Clinical Outcomes of COVID-19"

_medicina, 2021, doi:10.3390/medicina57090931_

Round 1
Reviewer 1 Report
This study assessed the association between aspirin and several COVID-19 outcomes. Overall, this paper is innovative and provided the evidence, but a few things have to be considered.
- For introduction part, there should be a citation for the cases and death numbers in Korea during the specific time period as well as the global numbers as of the date were investigated.
- The study introduction is weak and lack of established evidence discussion regarding the effectiveness of aspirin use in COVID. Several previous literatures demonstrated aspirin linked to less-severe COVID infections or reduce risk of death from COVID. Low-dose aspirin even can help prevent the infection from COVID. These evidences contrary to the findings in this study. Authors need to discuss why and give more explanation for the past literatures. The introduction is short and should be expanded with more details.
- The study assessed the participants were on aspirin within 2 weeks before diagnosis. This definition is tricky. Since they may also include the patients who were experiencing the COVID while they were not diagnosed. How will authors eliminate this situation?
- In the results part, the difference of composite endpoint 1 of COVID-19 diagnosis between Non-aspirin and Aspirin user before COVID-19 index date is marginal significance. Authors need to consider in expanding their sample size and see if this result is consistent.
- In the conclusion part, authors concluded that aspirin use before COVID-19 was related to an increased risk of worsened outcomes, especially death, and aspirin use after COVID-19 was related to an increased rate of conventional oxygen therapy. Did they also include the diseases or comorbidities of the patients who used aspirin before COVID-19?
- For the conclusion part, authors draw the conclusion that aspirin use was associated with adverse effects in COVID-19 patients. I am not confident to draw this conclusion. At least, authors should separate the situation with before and after diagnosis of COVID-19 since the before index involved a smaller number.
Author Response
Response to Reviewer 1 Comments
First of all, really thank you for your opinion.
Your advice was excellent and helpful to me.
Point 1: For introduction part, there should be a citation for the cases and death numbers in Korea during the specific time period as well as the global numbers as of the date were investigated.
Response 1: Thank you for your comment. I mentioned that.
Introduction 3~6 lines; More than 144,000 cases of COVID-19 and approximately 2000 deaths were reported throughout Korea between January 2020 and June 2021. Globally, to date, more than 100 million cases of COVID-19 and over 3 million deaths have been reported.
Point 2: The study introduction is weak and lack of established evidence discussion regarding the effectiveness of aspirin use in COVID. Several previous literatures demonstrated aspirin linked to less-severe COVID infections or reduce risk of death from COVID. Low-dose aspirin even can help prevent the infection from COVID. These evidences contrary to the findings in this study. Authors need to discuss why and give more explanation for the past literatures. The introduction is short and should be expanded with more details.
Response 2: Thank you for your comment. Now, in introduction section, the previous papers for aspirin with COVID were supplemented and I mentioned additional explanation with more detail. I made marking with red in article for additional explanation.
Point 3: The study assessed the participants were on aspirin within 2 weeks before diagnosis. This definition is tricky. Since they may also include the patients who were experiencing the COVID while they were not diagnosed. How will authors eliminate this situation?
Response 3: We used 2 weeks based on the previous references. Other papers for relationship with NSAID and COVID 19 defined medicine intake as more than 2 weeks.
Ex) PARK, Jungchan, et al. Non-steroidal anti-inflammatory agent use may not be associated with mortality of coronavirus disease 19. Scientific reports, 2021, 11.1: 1-7
Ex) SON, Minkook, et al. Effect of aspirin on coronavirus disease 2019: A nationwide case-control study in South Korea. Medicine, 2021, 100.30
Meanwhile, aspirin is rapidly absorbed in the upper gastrointestinal (GI) tract and results in a measurable inhibition of platelet function within 60 minutes. The plasma half-life of aspirin is only 20 minutes; however, because platelets cannot generate new COX, the effects of aspirin last for the duration of the life of the platelet (≈10 days). [AWTRY, Eric H.; LOSCALZO, Joseph. Aspirin. Circulation, 2000, 101.10: 1206-1218.] So, two weeks was suitable for duration to reflect the effect of aspirin.
Base cohort in our study were people who get COVID test. If people who are already experiencing COVID but have not been diagnosed, it could be false negative case. False negative results could be obtained for samples with low viral loads and those collected too early or too late during the clinical course of the disease and false negative rate in COVID 19 test was low such as 9.3%. [ KANJI, Jamil N., et al. False negative rate of COVID-19 PCR testing: a discordant testing analysis. Virology journal, 2021, 18.1: 1-6]
Failure to exclude false negative case is one of our limitations but it is hard to find the study that completely exclude limitations.
We totally agreed with you and we mentioned about this in limitations.
Point 4: In the results part, the difference of composite endpoint 1 of COVID-19 diagnosis between Non-aspirin and Aspirin user before COVID-19 index date is marginal significance. Authors need to consider in expanding their sample size and see if this result is consistent.
Response 4: Thank you for your valuable comments!!! Of course, it would be more accurate results with expanding sample size. However, unfortunately, the sample size cannot be increased arbitrarily because we should use the provided cohort from national health insurance system (NHIS). This cohort is strictly controlled by the NHIS and the number of patient for this cohort is fixed now.
Point 5: In the conclusion part, authors concluded that aspirin use before COVID-19 was related to an increased risk of worsened outcomes, especially death, and aspirin use after COVID-19 was related to an increased rate of conventional oxygen therapy. Did they also include the diseases or comorbidities of the patients who used aspirin before COVID-19?
Response 5: Thank you for your valuable comments!!! Among 22660 base cohort people, 1:1 propensity score matching was used to identify who took aspirin before diagnosis of covid and who had never taken it.
The Propensity core matching variables already included HTN, COPD, Asthma, CKD, DM, CVD, and Charlson combination index (0.1,2 or more).
Point 6: For the conclusion part, authors draw the conclusion that aspirin use was associated with adverse effects in COVID-19 patients. I am not confident to draw this conclusion. At least, authors should separate the situation with before and after diagnosis of COVID-19 since the before index involved a smaller number
Response 6: Thank you for your valuable comments!!! We rewrote the conclusion part.
; Aspirin use before the diagnosis of COVID tended to increase the death rate and aspirin use after the diagnosis of COVID tended to increase the conventional oxygen therapy rate than no use. Analysis with larger sample size is required to confirm the adverse effects of aspirin on COVID 19 and further research for mechanism is needed.

Reviewer 2 Report
AUTHORS
Title: Aspirin is related to worse clinical outcomes of COVID-19
This is a short but interesting study on the association between aspirin and COVID-19. Authors assessed outcomes by using nationwide data from the Korean National Health Insurance System, doing a retrospective observational cohort study that included patients who underwent COVID-19 testing in South Korea. I find the manuscript to be easy to read and interesting, compiling a large set of data but have many doubts on the experimental design.
A big question on the study is the robustness of the controls. Is aspirin an over the counter medicine in south korea? How are authors sure that controls did not take aspirin? You say on line 134 that aspirin exposed were matched to “non-aspirin-exposed patients”. How sure are authors that these “non-aspirin-exposed patients” indeed were not using aspirin?
Is there a possibility of bias on the aspirin-prescribed individuals? Could this drug be given more on those presenting more serious fevers (and hence with worst prognosis)? I believe this was not controlled and hence might also impact on your conclusion that “Aspirin use was associated with adverse effects in COVID-19 patients.”
Author Response
First of all, really thank you for your opinion.
Your advice was excellent and helpful to me.
Response to Reviewer 2 Comments
Point 1: A big question on the study is the robustness of the controls. Is aspirin an over the counter medicine in south korea? How are authors sure that controls did not take aspirin? You say on line 134 that aspirin exposed were matched to “non-aspirin-exposed patients”. How sure are authors that these “non-aspirin-exposed patients” indeed were not using aspirin?
Response 1: Thank you for your valuable comments!!! As you pointed out, actually, non-aspirin-exposed patients might contain aspirin users. Because we used only prescribed data though aspirin is over the counter medicine in Korea. Previous studies about the association between aspirin and COVID-19 also used only prescribed data. Your valuable comment will be the limitation of this study.
Ex) SON, Minkook, et al. Effect of aspirin on coronavirus disease 2019: A nationwide case-control study in South Korea. Medicine, 2021, 100.30.)
Ex) CHOW, Jonathan H., et al. Aspirin use is associated with decreased mechanical ventilation, intensive care unit admission, and in-hospital mortality in hospitalized patients with coronavirus disease 2019. Anesthesia & Analgesia, 2021, 132.4: 930-941
Ex) MERZON, Eugene, et al. The use of aspirin for primary prevention of cardiovascular disease is associated with a lower likelihood of COVID‐19 infection. The FEBS journal, 2021
Ex) Yuan S, Chen P, Li H, Chen C, Wang F, Wang DW. Mortality and prehospitalization
use of low-dose aspirin in COVID-19 patients with coronary artery disease. J Cell Mol Med 2020;25:1263–73.
Ex) Bachert C, Chuchalin AG, Eisebitt R, Netayzhenko VZ, Voelker M. Aspirin compared with acetaminophen in the treatment of fever and other symptoms of upper respiratory tract infection in adults: a multicenter, randomized, double-blind, double-dummy, placebo-controlled, parallel-group, single-dose, 6-hour dose-ranging study. Clin Ther. 2005 Jul;27(7):993-1003.
We could not find the study which included data about usage without prescription and we mentioned about this in limitations.
Point 2: Is there a possibility of bias on the aspirin-prescribed individuals? Could this drug be given more on those presenting more serious fevers (and hence with worst prognosis)? I believe this was not controlled and hence might also impact on your conclusion that “Aspirin use was associated with adverse effects in COVID-19 patients.”
Response 2: Because there is no specific guideline for the treatment of COVID-19 now. Aspirin is not preferred for high fever than acetaminophen. Acetaminophen and aspirin work to treat pain and fever by reducing inflammation in the body. Acetaminophen is generally used for pain and fever. Aspirin can also be used to prevent the risk of heart attacks and strokes in those with heart disease.
Aspirin may have more gastrointestinal side effects compared to acetaminophen so its use should be cautioned in those with a history of stomach ulcers. Acetaminophen, on the other hand, should be cautioned in those with liver disease, especially alcoholics.
Depending on patients’ condition and symptoms, one medication may be recommended over the other.
[Bachert C, Chuchalin AG, Eisebitt R, Netayzhenko VZ, Voelker M. Aspirin compared with acetaminophen in the treatment of fever and other symptoms of upper respiratory tract infection in adults: a multicenter, randomized, double-blind, double-dummy, placebo-controlled, parallel-group, single-dose, 6-hour dose-ranging study. Clin Ther. 2005 Jul;27(7):993-1003]
I agreed with your opinion and rewrote the conclusion part.
; Aspirin use before the diagnosis of COVID tended to increase the death rate and aspirin use after the diagnosis of COVID tended to increase the conventional oxygen therapy rate than no use. Analysis with larger sample size is required to confirm the adverse effects of aspirin on COVID 19 and further research for mechanism is needed.

Round 2
Reviewer 1 Report
Overall, this paper is much improved. I am satisfied with authors’ response and revision. Only one issue that authors need to address. As I suggested in the first version, authors should cite where they get the numbers of cases/death in Korean as well as the global numbers as of the date were investigated. The authors answered with “I mentioned that”. while “mentioned” does not equal to “evidence based”. They should cite where the numbers that got, like from CDC, WHO etc., especially for the first paragraph of the introduction.
Author Response
Response to Reviewer 1 Comments
Overall, this paper is much improved. I am satisfied with authors’ response and revision. Only one issue that authors need to address. As I suggested in the first version, authors should cite where they get the numbers of cases/death in Korean as well as the global numbers as of the date were investigated. The authors answered with “I mentioned that”. while “mentioned” does not equal to “evidence based”. They should cite where the numbers that got, like from CDC, WHO etc., especially for the first paragraph of the introduction.
Response : Thank you for your valuable comments.
I am really sorry. I misunderstood your advice.
I made cite for number of cases in Korea as well as Worldwide.
I made marking with red in new references [1] and [2]
Your advice was really helpful to me.
Thank you.
Reviewer 2 Report
All points raised were addressed. The manuscript seems solid and I advise publication
Author Response
Thank you for your valuable comments.
Your advice was really helpful to me.
Thank you.